# Single-Nucleotide Polymorphism in Genes Encoding G Protein Subunits *GNB3* and *GNAQ* Increase the Risk of Cardiovascular Morbidity among Patients Undergoing Renal Replacement Therapy

**DOI:** 10.3390/ijms242015260

**Published:** 2023-10-17

**Authors:** Simon Birkner, Birte Möhlendick, Benjamin Wilde, Kristina Schoenfelder, Kristina Boss, Winfried Siffert, Andreas Kribben, Justa Friebus-Kardash

**Affiliations:** 1Department of Nephrology, University of Duisburg-Essen, University Hospital Essen, Hufelandstr. 55, 45147 Essen, Germany; birkenersimon2@gmail.com (S.B.); benjamin.wilde@uk-essen.de (B.W.); kristina.schoenfelder@uk-essen.de (K.S.); kristina.boss@uk-essen.de (K.B.); andreas.kribben@uk-essen.de (A.K.); 2Institute of Pharmacogenetics, University Hospital Essen, University of Duisburg-Essen, 45147 Essen, Germanywinfried.siffert@uk-essen.de (W.S.)

**Keywords:** *GNB3* c.825C > T, *GNAQ* −695/−694GC > TT, *GNAS* c.393C > T, renal replacement therapy (RRT), cardiovascular events, left ventricular end-diastolic diameter (LVEDD), coronary artery stent insertion

## Abstract

Single-nucleotide polymorphisms in G protein subunits are linked to an increased risk of cardiovascular events among the general population. We assessed the effects of *GNB3* c.825C > T, *GNAQ* −695/−694GC > TT, and *GNAS* c.393C > T polymorphisms on the risk of cardiovascular events among 454 patients undergoing renal replacement therapy. The patients were followed up for a median of 4.5 years after the initiation of dialysis. Carriers of the TT/TT genotype of *GNAQ* required stenting because of coronary artery stenosis (*p* = 0.0009) and developed cardiovascular events involving more than one organ system (*p* = 0.03) significantly earlier and more frequently than did the GC/TT or GC/GC genotypes. Multivariate analysis found that the TT/TT genotype of *GNAQ* was an independent risk factor for coronary artery stenosis requiring stent (hazard ratio, 4.5; *p* = 0.001), cardiovascular events (hazard ratio, 1.93; *p* = 0.04) and cardiovascular events affecting multiple organs (hazard ratio, 4.9; *p* = 0.03). In the subgroup of male patients left ventricular dilatation with abnormally increased LVEDD values occurred significantly more frequently in TT genotypes of *GNB3* than in CT/CC genotypes (*p* = 0.007). Our findings suggest that male dialysis patients carrying the TT genotype of *GNB3* are at higher risk of left ventricular dilatation and that dialysis patients carrying the TT/TT genotype of *GNAQ* are prone to coronary artery stenosis and severe cardiovascular events.

## 1. Introduction

The risk of cardiovascular disease (CVD) is enormously increased in patients with kidney failure who are undergoing renal replacement therapy (RRT) [1]. Cardiovascular morbidity, with the occurrence of acute cardiovascular events, is the leading cause of mortality for dialysis patients, and their mortality rate is 20 times higher than that of the general population [1]. Multiple adverse clinical prognostic features are commonly accepted, such as age, male sex, duration and modality of dialysis treatment, and various dialysis-related effects that promote cardiovascular morbidity and mortality among patients undergoing RRT [1]. On the other hand, the results of a growing number of studies suggest that gene polymorphisms affect neurohormonal signaling pathways to magnify the risk of CVD among the general population. However, the specific population of patients undergoing RRT whose cardiovascular risk potential is already strongly elevated has not yet been investigated. Therefore, we regarded polymorphisms in genes encoding for G proteins mediating signaling cascades that control cardiovascular pathways as appealing candidates. 

The c.825C > T (rs5443) polymorphism of the guanine nucleotide-binding protein subunit β3 (*GNB3*), one of the genes that code for the β-subunits of G proteins, is the best described and most studied variant with functional consequences [2]. This single-nucleotide polymorphism results in an in-frame deletion of 123 nucleotides inducing the activation of a cryptic splice site; this activation leads to the formation of a truncated Gβ3-s protein [3,4]. Consequently, carriers of the TT genotype of *GNB3* exhibit only the modified Gβ3-s protein, which is responsible for the enhanced activation of G protein-induced signaling pathways [2,3,4]. The T allele has been shown to be associated with essential hypertension [2,4,5]. Carriers of the T allele were reported to develop left ventricular hypertrophy, independent of blood pressure and body composition [2,6,7]. This polymorphism in *GNB3* has been associated with an increased frequency of obesity, metabolic syndrome, and cardiovascular morbidity in many studies involving various ethnic groups [2,5,8,9]. Our previous work identified the c.825C > T polymorphism of *GNB3* as a relevant factor predicting the risk of cardiovascular mortality, in particular for new-onset myocardial infarction and acute peripheral artery occlusive disease (PAOD), among renal allograft recipients at 8 years follow-up after renal transplant [10].

Few reports have pointed out the direct effect of the *GNAQ* −695/−694GC > TT polymorphism on cardiovascular outcomes [11,12,13]. The human *GNAQ* gene encodes for the alpha subunit of the Gq protein. The GC to TT insertion–deletion is a common variation in the *GNAQ* gene [11]. The dinucleotide substitution occurs in the promoter region of *GNAQ* and alters both promoter activity and mRNA production of Gαq [11]. The data on the functional effects of the *GNAQ* promoter polymorphism are controversial [11,12,13]. Liggett et al. found that the TT/TT variant is linked to increased promoter activity in cultured rat cardiac myocytes and human fibroblasts [11]. Thus, the *GNAQ* −695/−694GC > TT polymorphism induces the expression of Gαq [11]. The TT/TT genotype has been associated with poorer survival rates among African Americans with heart failure [11]. On the other hand, Frey and colleagues reported an augmented signal transduction via Gq-coupled angiotensin II receptor type 1, attributed to a stronger binding of the GC variant to the *GNAQ* promoter [12,13]. In contrast to Liggett’s observation, another study found that Gαq expression in human myocytes is higher among carriers of the GC/TT or GC/GC genotype than among carriers of the TT/TT genotype [13]. The vasocontractile response to angiotensin II stimulation is also stronger in the skin and saphenous veins of GC/TT or GC/GC genotype carriers than in those of TT/TT genotype carriers [13]. The increase in troponin I levels after coronary artery bypass grafting was higher among carriers of the GC/TT or GC/GC genotype than among carriers of the TT/TT genotype [12]. Furthermore, Frey et al. observed an elevated prevalence of cardiac hypertrophy among carriers of the GC/TT or GC/GC genotype [12].

Another gene encoding for the α-subunit of G proteins is *GNAS*. The single-nucleotide polymorphism c.393C > T (rs7121) of the *GNAS* gene induces altered mRNA folding, which affects mRNA stability and subsequently evokes increased expression of mRNA and of Gαs protein among subjects with the T allele [14,15]. Increases both in the release of Gαs and in cAMP synthesis were observed in vitro in cardiomyocytes from T allele carriers, and these increases resulted in increased contractile capability [14,15]. Moreover, previous studies found a positive association between the T allele variant of the *GNAS* c.393C > T polymorphism and the risk of essential hypertension, ventricular tachyarrhythmias, sudden cardiac death, and reduced response to beta-blocker therapy, a finding supporting the potential relevance of this polymorphism for cardiovascular morbidity [16,17].

We hypothesized that these three functionally relevant single-nucleotide polymorphisms in the three candidate genes described above may influence the susceptibility of patients undergoing RRT to alterations of echocardiographic parameters of the left heart and cardiovascular events and may also be a useful predictor of the risk of these outcomes for RRT patients. 

## 2. Results

### 2.1. Patient Characteristics

Our study cohort consisted of 454 patients with kidney failure who were undergoing RRT. The patients applied between March 2007 and February 2022 to the waiting list for a renal allograft at the University Hospital Essen and were retrospectively followed up for a median of 4 years after the initiation of RRT. As summarized in Table 1, the median age of the cohort was 51 years (range, 18 to 81 years). A total of 168 women (37%) and 286 men (63%) were included. Of the study cohort, 96 patients (21%) were undergoing peritoneal dialysis; 72 (16%) had undergone a previous kidney transplant; 115 (25%) were obese; and 69 (15%) had diabetes mellitus type I, II, or III at baseline (Table 1). The most common causes of renal failure in the cohort were glomerulonephritis, nephrosclerosis, and polycystic kidney disease (Table 1). 

The distribution of the three relevant single-nucleotide polymorphisms encoding G protein subunits is shown in Table 1. The distribution of the genotypes of the three polymorphisms corresponded to the Hardy–Weinberg equilibrium. With regard to the basic demographic characteristics, the causes of renal failure, and the presence of diseases and factors influencing the risk of cardiovascular events, we observed no significant differences between homozygous T allele carriers and C allele carriers of any of the three genes (*GNB3*, *GNAQ*, and *GNAS*; Table 1). The only notable difference was a higher proportion of women among carriers of the TT/TT genotype of *GNAQ* (Table 1).

Taking into account the entire cohort, cardiovascular events appeared in 69 of 454 patients (15%) undergoing RRT (Table 2). Only 14 patients (3%) experienced cardiovascular events in multiple organ systems (Table 2). During follow-up, 34 patients (7%) required coronary artery stent placement, and 25 (5%) patients experienced a myocardial infarction (Table 2). Coronary artery bypass grafting was required for 14 (3%) patients (Table 2). Acute PAOD occurred among 18 patients (4%), and stroke occurred among 13 patients (3%) (Table 2). New-onset chronic cardiovascular disease, such as coronary artery disease (CAD), occurred during dialysis among 60 patients (13%), and new-onset heart failure occurred among 68 (15%) of the patients undergoing RRT (Table 2). Left ventricular dilatation with left ventricular end-diastolic diameter (LVEDD) above normal values was detected in 78 patients in the study cohort (Table 2).

### 2.2. For Patients Undergoing RRT, the TT/TT Genotype of GNAQ Confers an Increased Risk of the Need for Coronary Artery Stenosis Requiring Stent Insertion and of the Occurrence of Cardiovascular Events, Particularly Those Involving Multiple Organ Systems

There is little evidence to support the potential influence of the *GNAQ* −695/−694 GC > TT polymorphism on the risk of cardiovascular disease (11–13). Therefore, we wondered whether the presence of a homozygous genotype of this polymorphism in *GNAQ* might increase the already elevated risk of cardiovascular events among patients with RRT. First, we found that carriers of the TT/TT genotype required coronary artery stent insertion due to the coronary artery stenosis significantly more often during RRT than did carriers of the GC/TT or GC/GC genotype (14 (14%) vs. 20 (6%); *p* = 0.003; Table 2). Survival without coronary artery stenosis requiring insertion of a coronary artery stent was significantly better among carriers of the GC/TT or GC/GC genotype than among carriers of the TT/TT genotype (*p* = 0.0009; Figure 1A). In addition, the number of patients who experienced new-onset coronary artery disease during RRT was slightly higher among carriers of the TT variant of *GNAQ* (18 (19%) vs. 42 (12%); *p* = 0.08; Table 2). There was a trend toward an increased frequency of cardiovascular events during RRT in the TT/TT genotype carriers (20 (21%) vs. 49 (14%); *p* = 0.09; Table 2). Survival rates without a cardiovascular event tended to be worse among patients with the TT/TT genotype (*p* = 0.057; Figure 1B). Interestingly, dialysis patients who experienced cardiovascular events in multiple organ systems during RRT were significantly more likely to exhibit the TT/TT genotype (6 (6%) vs. 8 (2%); *p* = 0.05; Table 2). Consequently, the presence of the TT variant of *GNAQ* had a negative impact on the risk of the development of cardiovascular events involving more than one organ system during dialysis treatment (*p* = 0.03; Figure 1C). After the Bonferroni correction as well as the Benjamini and Hochberg correction in Table 2 only the association of the occurrence of coronary artery stenosis requiring stent insertion with the TT/TT genotype of *GNAQ* remained statistically significant.

Interestingly, further analysis of cardiovascular morbidity in age-related groups found that patients carrying the TT/TT genotype of *GNAQ* and being 18 to 39 years old developed significantly more frequently cardiovascular events, in particular involving more than one organ system, coronary artery stenosis requiring stenting and major cardiovascular events as well as new onset coronary artery disease during the RRT comparing to the GC/TT and GC/GC genotypes of *GNAQ* of the same age (Appendix A). In addition, coronary artery stent insertion because of coronary artery stenosis was required significantly more often in the group of TT/TT genotypes of *GNAQ* who were 50 to 59 years old than in the group of GC/TT/ and GC/GC genotypes of the same age (Appendix A). Our results suggest that the TT/TT genotype of *GNAQ* exerts a negative effect on cardiovascular morbidity in young patients undergoing RRT. In older dialysis patients with an already strongly increased cardiovascular risk due to age, the TT/TT genotype of *GNAQ* did not expose additional negative impact on cardiovascular morbidity and plays a minor role compared to the age factor. Sex stratification revealed that stent insertion due to coronary artery stenosis was needed significantly more frequently in TT/TT genotype carriers of *GNAQ* than in GC/TT or GC/GC genotype carriers in the male as well as in the female population of our study cohort, while cardiovascular events, in particular cardiovascular events involving multiple organ systems and acute PAOD appeared during RRT significantly more often in TT/TT genotype carriers than in those carrying the GC/TT or GC/GC genotype only the subgroup of females (Appendix A). In the female subpopulation, TT/TT genotypes of *GNAQ* developed coronary artery disease significantly more frequently during RRT than the GC/TT or GC/GC genotypes (Appendix A).

Next, we evaluated the effect of non-genetic factors other than the above-mentioned *GNAQ* polymorphism that may modify the risk of the development of cardiovascular events, especially those involving multiple organ systems, and the risk of the need for coronary artery stent placement due to coronary artery stenosis during RRT. The univariate analysis found that the non-genetic variables, older age, male sex, prolonged time on RRT, preexisting diabetes mellitus, and abnormally elevated LVEDD reflecting the occurrence of left ventricular dilatation during RRT are potential modifiers of the risk of coronary artery stenosis requiring a coronary artery stent during dialysis (Table 3). The subsequent multivariable Cox regression showed that the TT/TT genotype of the *GNAQ* −695/−694GC > TT polymorphism, the history of diabetes mellitus, and age were independent risk factors for coronary artery stenosis with the need for coronary artery stent insertion during RRT (Table 3). With regard to the development of cardiovascular events during RRT follow-up, the previous univariate analysis determined that, in addition to the TT/TT genotype of *GNAQ* as a genetic factor, several non-genetic factors, older age, male sex, longer time on RRT, and preexisting diabetes mellitus negatively affected the risk of the occurrence of cardiovascular events during dialysis (Table 4). Multivariable analysis showed that older age and preexisting diabetes mellitus were relevant parameters contributing to the increased risk of the development of cardiovascular events during RRT, whereas the presence of the TT/TT genotype of *GNAQ* exerted a moderate effect (Table 4). The results of the univariate analysis for the risk of the development of cardiovascular events involving multiple organ systems indicated that most patients who experienced cardiovascular events affecting more than one organ system were TT/TT genotype carriers and also were older, spent a longer time on dialysis, had undergone a previous kidney transplant, and displayed LVEDD values above the norm (Table 5). Data obtained from additional Cox regression analyses demonstrated that older age, duration of RRT, obesity, LVEDD deviation from the upper limit of normal, and TT/TT genotype of *GNAQ* were strong factors negatively mediating the risk of the occurrence of cardiovascular events involving multiple organs at the time of RRT (Table 5).

### 2.3. The TT Genotype of GNB3 Is an Independent Risk Factor for Abnormally Increased LVEDD in Male Dialysis Patients

The c.825C > T single-nucleotide polymorphism of *GNB3* has been widely studied, and many published reports have shown that the TT genotype confers an increased risk of arterial hypertension, development of left ventricular hypotrophy, and occurrence of common acute cardiovascular events such as myocardial infarction or stroke among populations without chronic kidney disease [2,4,5,6,7,8,9,10]. Therefore, our study was designed to examine the causal relationship between the TT genotype of *GNB3* and the development of cardiovascular events and alterations of the echocardiographic parameters of the left ventricle in the special population of patients undergoing RRT who are strongly prone to cardiovascular diseases. As indicated in Table 2, the proportions of patients in whom cardiovascular events developed were comparable between the TT genotype and the CT/CT genotypes. We also found no significant differences between the genotypes with regard to various types of cardiovascular events or the occurrence of cardiovascular events involving multiple organ systems (Table 2). With regard to variations in certain echocardiographic parameters, we also did not detect any significant association with the TT genotype of *GNB3* (Table 2). Similarly, the rates of survival without acute cardiovascular events did not differ between homozygous T allele carriers and non-carriers (*p* = 0.49). 

As shown in Appendix A, regarding the entire study cohort, a significant increase in LVEDD values occurred in patients with the TT genotype of *GNB3* compared to the CT/CC genotypes only for patients who were 50–59 years old. Considering other age groups, no differences between the *GNB3* TT genotype carriers and those with the CT/CC genotypes of *GNB3* were identified (Appendix A). Thus, the association between the abnormal increase in LVEDD indicating left ventricular dilatation and the TT genotype of *GNB3* seems to be relevant only in elderly patients. On the other hand, the occurrence of LVEDD increase in patients who were 50–59 years old might be predominantly attributed to the high age.

However, previous reports have described sex-specific effects of the c.825C > T single-nucleotide polymorphism of *GNB3*, showing that the TT genotype is linked to poorer outcomes of cardiovascular disease among male subjects [18]. Therefore, in this study of dialysis patients, we stratified the analysis for gender. This sex stratification resulted in an increased number of patients with abnormally elevated LVEDD indicative of left ventricular dilatation among male patients carrying the TT genotype compared to those carrying the CT/CC genotypes (10 (50%) vs. 39 (26%); *p* = 0.02; Table 6). The further Bonferroni correction as well as the Benjamini and Hochberg correction performed in Table 6 attenuated the statistically significant link between the TT genotype of *GNB3* and abnormal increase in LVEDD values in male subjects. Conversely, among female patients, we observed a trend toward a protective effect of the TT genotype with respect to the development of increased LVEDD values (0 (0%) vs. 29 (30%); *p* = 0.054; Table 6). Stratification by sex did not detect any difference in the risk of cardiovascular events according to the various *GNB3* genotypes (Table 6). Although we found no differences in the entire study cohort between the *GNB3* genotypes in rates of survival free of an increase in LVEDD, (*p* = 0.16; Figure 2A), the stratified analysis found better abnormal LVEDD increase-free survival rates among male patients carrying the CT/CC genotypes than for those carrying the TT genotype (*p* = 0.007; Figure 2B). Univariate analysis, performed to assess which other non-genetic factors might influence the risk of the development of an abnormal increase in LVEDD among male patients, found that the TT genotype of the *GNB3* polymorphism was the only variable with a significant effect on the occurrence of an abnormal increase in LVEDD value corresponding to the left ventricular dilatation during RRT (Table 7). A Cox regression analysis adjusted for all other potential modifiers of the risk of abnormal increase in LVEDD corresponding to left ventricular dilatation found that the TT genotype of *GNB3* was still an independent risk factor (Table 7).

It is worth noting that an analysis of the changes in antihypertensive medication at RRT follow-up found that, among the entire study population of dialysis patients, the proportion of patients who received an enhancement in antihypertensive therapy was significantly higher among TT genotype carriers than among non-carriers (9 (21%) vs. 36 (9%); *p* = 0.02; Table 8). When we focused on the male subgroup, these effects disappeared (2 (7%) vs. 25 (10%); *p* = 0.57; Table 8). In fact, among the female population, significantly more TT genotype carriers than CT/CC genotype carriers required an intensification of antihypertensive therapy during RRT (7 (50%) vs. 11 (7%); *p* ≤ 0.0001; Table 8). This observation led us to the assumption that adjustments in antihypertensive therapy among TT genotype carriers in the entire cohort and especially among women might explain the lack of the expected higher prevalence of left ventricular dilatation with increased LVEDD values among TT genotypes in the entire study cohort or among female patients. Otherwise, antihypertensive treatment was not intensified in male dialysis patients with the TT genotype. In this way, this high-risk group of male TT genotype carriers was not protected from the development of left ventricular dilatation with abnormally increased LVEDD values due to the TT genotype of *GNB3*. 

### 2.4. The GNAS c.393C > T Polymorphism Is Not Associated with Cardiovascular Events or Abnormal Increase in LVEDD among Patients Undergoing RRT

With regard to the *GNAS* c.393C > T polymorphism, we observed no *GNAS* genotype-dependent differences in the development of cardiovascular events or in the occurrence of abnormally increased LVEDD values among patients undergoing RRT (Table 2). We further saw no differences between the two genotypes in different age groups according to the performed age stratification (Appendix A). There were also no significant differences in cardiovascular outcomes attributed to sex (Appendix A).

## 3. Discussion

The present study, which involved a cohort of dialysis patients with a high preexisting risk of cardiovascular events, was conducted to investigate the effect on cardiovascular risk of the functional single-nucleotide polymorphisms in three candidate genes that are involved in G protein-mediated signaling pathways by activating the angiotensin II and beta-adrenergic receptors. Our population of patients undergoing RRT exhibited an association between the presence of the TT/TT genotype of the *GNAQ* promotor polymorphism and the occurrence of cardiovascular events, in particular coronary artery stenosis requiring stent placement and the occurrence of multiple cardiovascular events involving multiple organ systems. Surprisingly, for c.825C > T, the common functional polymorphism in *GNB3*, there were no differences between the genotypes with respect to the occurrence of cardiovascular events or the development of an abnormal increase in LVEDD due to left ventricular dilatation. However, male patients carrying the TT genotype of *GNB3* exhibited a significantly higher incidence of abnormally elevated LVEDD at the echocardiographic examination performed at the end of follow-up. Intensification of antihypertensive therapy at RRT follow-up was significantly more frequent among the subgroup of female patients with the TT genotype. For the third polymorphism, *GNAS* c.393C > T, no significant associations with cardiovascular risk were observed among patients undergoing RRT. 

Recent studies have shown a clear correlation between arterial hypertension and the T allele variant of the *GNB3* c.825C > T polymorphism; these results were attributed to the increased activation of G protein-coupled signal transduction in the subgroup of white patients [2,4,5,19]. In our study population of dialysis patients, the proportion of patients who required intensification of antihypertensive treatment during RRT was significantly higher among carriers of the TT genotype than among carriers of the CT/CT genotypes of *GNB3*. We also detected a slight trend toward fewer patients with the TT genotype who were being treated with fewer than five antihypertensive drugs or who required no adjustment in the dosage of antihypertensive medication. The effect of the TT genotype on the need for antihypertensive drug escalation was particularly pronounced among women. Conversely, compared to the female TT genotype carriers, the female CT/TT genotype carriers were significantly more frequently treated with a low number of antihypertensive drugs and without changing the treatment regimens. These observations are consistent with recently published results. 

However, for patients with impaired renal function, some published results contradict our findings [20,21]. Wüthricht et al. found that, among 216 renal allograft recipients during the first year after transplant, the T allele exerted no negative effect on arterial hypertension or the need for adjustments in the antihypertensive medication regimen [21]. The discrepancy between these results and our findings may be related to age-specific effects, the severity of renal dysfunction, the promoting effect of certain immunosuppressive agents, or the shorter follow-up period [21]. Schunkert et al. postulated that the effect of the *GNB3* c.825C > T polymorphism is restricted to older patients with low-renin essential hypertension [22]. Moreover, the development of arterial hypertension, especially under conditions of deteriorating renal function, is a multifactorial process in which the influence of environmental factors may outweigh the mild effects of genetic determinants. 

Numerous studies have ruled out the *GNB3* c.825C > T polymorphism as an essential risk factor for the development of left ventricular hypertrophy [2,6,7]. Poch et al. suggested a relationship between the T allele and the risk of left ventricular hypertrophy among patients with a history of arterial hypertension [6]. Strikingly, as illustrated by Semplicini et al., the association of T allele carrier status with left ventricular growth persisted for young normotensive patients [7]. The work of Mahmood et al. reinforced the observation that, among T allele carriers in the Emirati population, left ventricular hypertrophy occurred independently of the development of essential hypertension [23]. These findings may provide evidence that the T allele has a direct effect on the growth processes of the left ventricle, independent of blood pressure elevation [23]. A possible mechanism by which the TT genotype of the *GNB3* c.825C > T polymorphism may be involved in cardiomyocyte proliferation is the increased ability of the Na/H exchange to alter intracellular pH as a result of the overactivation of G protein-coupled receptors because of the presence of the T allele [23,24]. Nevertheless, we observed no difference between the *GNB3* genotypes in our study population with regard to echocardiographic parameters characterizing left ventricular hypertrophy and dilatation as a possible consequence of left ventricular hypertrophy.

Of note, some previous reports have mentioned sex-specific differences in the phenotypic expression of the *GNB3* c.825C > T polymorphism [18,25]. Frey et al. described sex-specific effects of the *GNB3* c.825C > T polymorphism on the risk of myocardial infarction; the TT genotype was detrimental for male subjects [18]. Furthermore, Gbadoe et al. showed increased levels of soluble selectin among male patients carrying the T allele that was associated with an increased risk of inferior outcome of cardiovascular disease among men but not among women [25]. Hence, we separately analyzed surrogate variables of left ventricular dilatation and hypertrophy as detected by echocardiographic examination of women and men. Interestingly, among male patients, the TT genotype was an independent predictor of abnormally elevated LVEDD values corresponding with the development of left ventricular dilatation that might be a result of the previous left ventricular hypertrophy. Among female patients, LVEDD did not differ between carriers of the *GNB3* genotype. A possible reason that the T allele does not lead to left ventricular dilatation among female patients may be that female carriers of the TT genotype in our study cohort population required modification of antihypertensive drug therapy significantly more frequently than did female CT/CC genotype carriers, and this difference may have rescued them from the progression of left ventricular proliferation and development of left ventricular dilatation as a consequence. Furthermore, a significantly more intensive antihypertensive treatment regimen for TT genotypes than for CT/CC genotypes may suggest attenuation of the T allele effect on the LVEDD elevation in the entire study cohort.

In our study, patients carrying the TT genotype were not at higher risk of cardiovascular events or new-onset coronary artery disease. The results of subsequent separate analyses of male and female subgroups confirmed these observations. It should be noted that previous studies examining the association between the T variant of *GNB3* and cardiovascular morbidity and mortality have produced inconsistent results [5,9,10,18,26,27,28,29,30,31]. Several case–control and longitudinal studies have failed to demonstrate an increased risk of cardiovascular mortality and morbidity among T allele carriers in white populations [28,29,30,31], whereas other studies have demonstrated this increased risk [5,9,10,18,26,27]. Differences in study design may underlie these conflicting results. Most of the studies that have found a higher risk have involved a follow-up of more than 5 years. Notably, the median follow-up in our study was only 4 years. It is conceivable that an extended follow-up period could capture more cardiovascular events and detect differences between the *GNB3* genotypes.

In terms of the dinucleotide exchange polymorphism in the *GNAQ* promoter, −695/−694 GC > TT, we found evidence of a higher risk of cardiovascular events, particularly in multiple organs, and of the need for a coronary artery stent among dialysis patients carrying the TT/TT genotype of *GNAQ* but not among those carrying the GC/TT or GC/GC genotype. However, the risk of left ventricular dilatation with a corresponding increase in LVEDD was not different among TT/TT genotype carriers. These findings could be reconciled with the report of Liggett et al. [11], whose data indicated stronger binding of the Sp-1 protein to the promoter of Gαq, resulting in augmented promoter activity for the TT variant of *GNAQ* at baseline and after stimulation with phorbol ester in vitro by applying cultured neonatal rat cardiac myocytes and human embryonic kidney (HEK) fibroblasts [11]. Gαq plays a central role in myocardial hypertrophy [32]. Downstream pathways mediated by the angiotensin II receptor type 1 are involved in myocardial hypertrophy [33]. Mice overexpressing the angiotensin II receptor suffered from hypertrophy and heart failure [33,34]. The angiotensin II receptor type 1 is a classic G protein-coupled receptor that transduces its signal through Gαq-induced pathways [32,33]. Blocking Gαq or knocking out the genes responsible for Gαq production can protect mice against the development of cardiac hypertrophy and heart failure [35,36]. In addition to cardiac hypertrophy, coronary vasoconstriction occurs in response to the activation of the Gαq-induced signaling cascade by angiotensin II which activates phospholipase c beta and ends in intracellular calcium influx [37]. Therefore, increased expression of Gαq associated with the TT/TT genotype of *GNAQ* may initiate coronary artery constriction, predisposing the patient to coronary artery stenosis requiring stent insertion and to severe cardiovascular events, as shown in our analysis of TT/TT genotype carriers. However, Liggitt et al. observed no association between the TT/TT genotype and left ventricular hypertrophy or death from heart failure in the white population [11]. In fact, African Americans with the TT/TT genotype were more likely to die of heart failure than were white patients [11]. In agreement with Liggett and colleagues, our study did not show an association between the TT/TT genotype and left ventricular dilatation with an abnormal LVEDD increase detected by echocardiography. However, most of the available findings about the linkage between the *GNAQ* promoter polymorphism of Gαq and left ventricular growth and vasoconstriction processes were derived from in vitro and murine experiments and cannot always be extrapolated to human studies.

The research group of Frey et al. provided a completely opposite insight into Gαq expression due to the *GNAQ* −695/−694 GC > TT polymorphism, suggesting that the enhanced transcriptional activation of Gαq by angiotensin II relied on the GC allele [12,13]. In contrast to Liggitt’s and our results, subjects carrying the GC/TT or GC/GC genotype were rendered more susceptible to left ventricular hypertrophy than those carrying the TT/TT genotype [12]. In vitro experiments by Frey et al. also revealed increased vasoconstriction in response to angiotensin II stimulation [13]. It is generally known that patients with an impaired renal function who are undergoing RRT exhibit chronic activation of the renin–angiotensin–aldosterone axis with elevated levels of circulating angiotensin II and volume overload. Therefore, the use of angiotensin II receptor antagonists or angiotensin-converting enzyme inhibitors is a common treatment approach for this special population. Assuming that the observations of Frey et al. are correct and that carriers of the GC/TT or GC/GC genotypes express more Gαq than non-carriers, we could speculate that the use of antihypertensive therapy that inhibits angiotensin II-induced signaling through Gαq might abolish the adverse effects of GC/TT or GC/GC genotype status on cardiovascular morbidity in our cohort of dialysis patients. Perhaps the GC/TT or GC/GC genotype carriers may be more sensible to angiotensin II receptor blockade with appropriate antihypertensive drugs because of the chronic setting of Gαq overexpression, or they may receive higher doses of these drugs, thus minimizing their risk of cardiovascular events and coronary artery stenosis, arguing for the higher cardiovascular risk among TT/TT genotype carriers than among GC/TT or GC/GC genotype carriers observed in the current study. Otherwise, in murine studies, Gαq overexpression switched on anti-apoptotic pathways in cardiomyocytes that may be meaningful for the protection of GC/TT or GC/GC genotype carriers from cardiovascular events [38]. In conclusion, additional studies are needed to evaluate Gαq production in different *GNAQ* genotypes and to elucidate its association with cardiovascular morbidity and mortality in various at-risk populations. 

Despite the already known Gαs activation in T allele carriers of the *GNAS* c.393C > T polymorphism, the present study found no difference in cardiovascular morbidity between T allele carriers and non-carriers [14,15]. Although the hypothesis of an association between Gαs overexpression in TT genotypes and negative effects on vasoconstriction or left ventricular growth seems to be pathomechanistically plausible, published reports on the association of *GNAS* with cardiovascular risk are rare [16,17].

Our study has several limitations, including a relatively short follow-up of a median of 4.5 years and a retrospective design. Mainly due to the single-center design of the study the number of patients and in particular the corresponding cardiovascular events were not so large, and we cannot fully exclude the observed associations with cardiovascular morbidity after testing the three different single-nucleotide polymorphisms acting in the same G protein-mediated signal pathways for multiple cardiovascular events might be partly caused by random chance. It is a common practice in our center that we did not recommend an application for renal transplant for those patients with very high cardiovascular morbidity that would not survive the transplantation surgery. Hence, those patients under RRT who have a very high risk for cardiovascular events are lacking in our study cohort which might be a selection bias in our study. Another limitation of the present study is the small number of cardiovascular events resulting in relatively small numbers in the subgroups used for multivariable models and contributing to a preliminary nature of the results of multivariable analyses. Echocardiographic examination and information about changes in antihypertensive therapy were missing for some patients, and their absence may influence our results on the development of left ventricular dilatation, and this is another limitation of the study. We were not able to assess the development of left ventricular hypertrophy in the study because the echocardiographic parameter measurements related to the hypertrophy were available only in a few patients of the cohort. After the Bonferroni correction only, the association between the development of coronary artery stenosis requiring stenting during RRT and the TT/TT genotype of GNAQ remained statistically significant. However, the term coronary artery stenosis requiring stenting united in itself acute myocardial infarctions as acute coronary artery disease and chronic coronary artery disease containing persistent angina pectoris. To our knowledge, this is the first study to investigate the relevance of functional polymorphisms in G protein-coding genes for cardiovascular morbidity in patients undergoing RRT. In conclusion, our results suggest that homozygous T allele carriers of the *GNB3* polymorphism who are undergoing RRT, especially men, may benefit from intensive antihypertensive therapy to avoid the rapid manifestation of left ventricular hypertrophy, whereas, for the TT/TT genotypes of the *GNAQ* promoter polymorphism in dialysis patients, close clinical control and strict lifestyle modification and treatment aimed at reducing the risk of coronary artery stenosis and acute cardiovascular events may be recommended. 

## 4. Materials and Methods

### 4.1. Study Population

This retrospective single-center study investigated the relationship between the three polymorphisms in genes encoding G protein subunits, *GNB3* c.825C > T, *GNAQ* −695/−694 GC > TT, and *GNAS* 393C > T. The study involved 454 patients undergoing RRT, who all applied for a waiting list between March 2007 and February 2022 at the University Hospital Essen. After the application, our center screened the patients and evaluated whether the patients were eligible for renal transplantation. Dialysis patients younger than 18 years were excluded from the study cohort. The study protocol was approved by the Ethics Committee of the University Hospital Essen (22–10546-BO). Of the total cohort, 358 (79%) were undergoing hemodialysis, and 96 (21%) were undergoing peritoneal dialysis. Samples of whole blood were collected once when patients applied for inclusion on the waiting list. Clinical data were obtained by a review of medical records. Patients were followed up for a median of 4.5 years from the start of RRT until April 2023.

Clinical outcome variables were the occurrence of cardiovascular events during dialysis and the development of abnormalities in available echocardiographic parameters of the left ventricle during RRT. Instances of cardiovascular events and newly diagnosed coronary artery disease or heart failure during RRT were obtained from each patient’s medical history and are listed in Table 2. We took into account for our analysis only incident cardiovascular events that occurred after the initiation of RRT. The values of the echocardiographic parameters of the left ventricle such as LVEDD, interventricular septal end diastole (IVSd), and left ventricular posterior wall end diastole (LVPWd) were obtained from echocardiographic measurements made at the end of the RRT follow-up period. Normal values for these echocardiographic variables were set in accordance with the recommendations of the British Society of Echocardiography guidelines [39].

The diagnosis of acute myocardial infarction was based on typical electrocardiographic changes and an increase in the activity of cardiac enzymes. Only PAOD requiring arterial revascularization by either open surgery or an endovascular intervention was defined as an acute PAOD event. Acute PAOD, transient ischemic attack, stroke, carotid artery stenosis, myocardial infarction, coronary artery stenosis requiring stent insertion, and coronary artery bypass surgery were included in the definition of cardiovascular events. Cardiovascular death, stroke, and myocardial infarction were defined as the major adverse cardiovascular events (MACE). All cardiovascular events were subclassified as those affecting the brain (transient ischemic attack, stroke, carotid artery stenosis), those affecting the heart (myocardial infarction, coronary artery stenosis with stent insertion, coronary artery bypass grafting), and those affecting the limbs (acute PAOD). These events were defined as cardiovascular events involving multiple organ systems. Information about therapy with antihypertensive and lipid-lowering drugs was obtained from medical reports generated at the initiation of RRT. Changes in antihypertensive drug therapy were obtained from medical reports generated at the end of follow-up.

### 4.2. Genotyping for GNB3, GNAQ, and GNAS

Genomic DNA was extracted from 200 µL EDTA-treated blood with the QIAamp DNA Blood Mini Kit (Qiagen, Hilden, Germany). Polymerase chain reaction (PCR) was performed with 2 µL genomic DNA and 30 µL Taq DNA-Polymerase 2x Master Mix Red (Ampliqon, Odense, Denmark) under the following conditions: initial denaturation at 94 °C for 3 min; 38 cycles with denaturation at 94 °C for 30 s, annealing at 62–66 °C for 30 s, and elongation at 72 °C for 30 s each; and final elongation at 72 °C for 10 min. Primers, annealing temperatures, and the respective restriction enzymes are presented in Table 9. For the various genotypes, the results of restriction fragment length polymorphism (RFLP)-PCR were validated by Sanger sequencing. Hardy–Weinberg equilibrium (HWE) was calculated with Pearson’s chi-square (*χ*^2^) goodness-of-fit test, and samples were considered to be deviant from HWE at a significance level of *p* ≤ 0.05. Genotypes for *GNAS* rs7121 (*χ*^2^ = 0.06; *p* = 0.80), *GNAQ* rs72466452 (*χ*^2^ = 2.78; *p* = 0.10), and *GNB3* rs5443 (*χ*^2^ = 0.002; *p* = 0.99) were compatible with HWE.

### 4.3. Statistical Analysis

Categorical variables were expressed as numbers and percentages and were compared using the two-tailed *χ*^2^ test. Significant differences between continuous variables were analyzed with the Mann–Whitney test. Survival curves were generated by the Kaplan–Meier method, and comparisons of survival rates were conducted with the log-rank test. The multivariable Cox regression analysis was adjusted for selected covariates that had reached a significance level of *p* = 0.1 in a previous univariate analysis. For the remaining analyses, statistical significance was set at the level of *p* < 0.05. Statistical analyses were calculated with GraphPad Prism version 6 (GraphPad Software, Inc., La Jolla, CA, USA) and IBM SPSS Statistics version 23 (IBM Corp., Armonk, NY, USA).

## Figures and Tables

**Figure 1 ijms-24-15260-f001:**
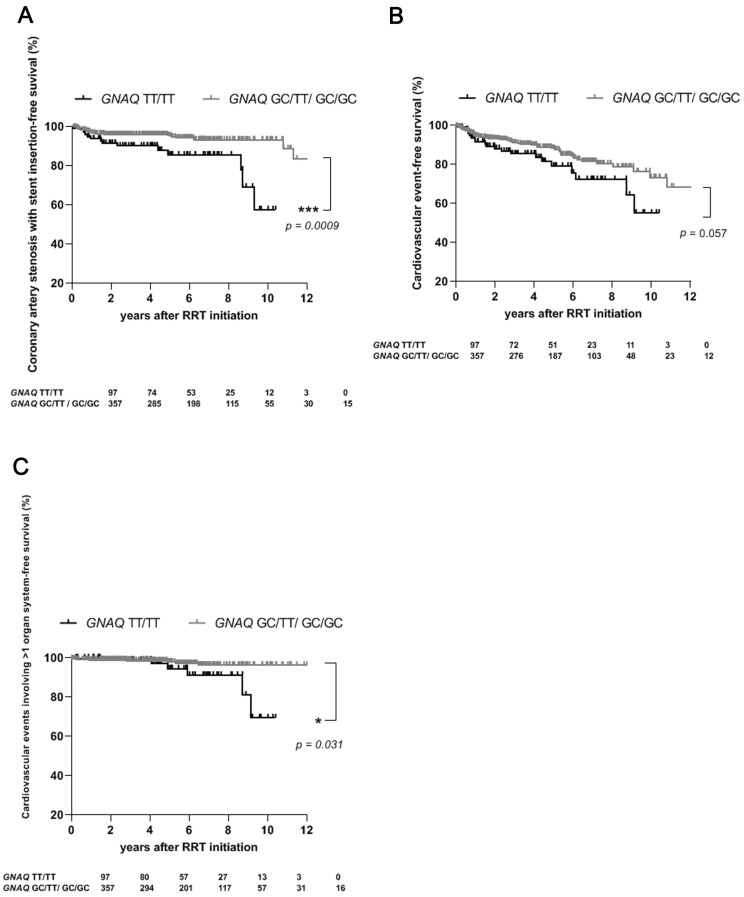
Occurrence of cardiovascular events among carriers of the TT/TT genotype and carriers of the GC/TT or GC/GC genotype of *GNAQ* during the follow-up period of renal replacement therapy (RRT). (**A**) Survival dependent on the appearance of coronary artery stenosis requiring stent insertion among carriers of the TT/TT or the GC/TT/GC/GC genotype of *GNAQ* (*p* = 0.0009). (**B**) Survival dependent on the occurrence of a cardiovascular event according to the genotype of *GNAQ* (*p* = 0.057). (**C**) Survival dependent on the occurrence of cardiovascular events involving more than one organ system according to the genotype of *GNAQ* (*p* = 0.03). *, *p* = 0.05; ***, *p* = 0.001. RRT, renal replacement therapy.

**Figure 2 ijms-24-15260-f002:**
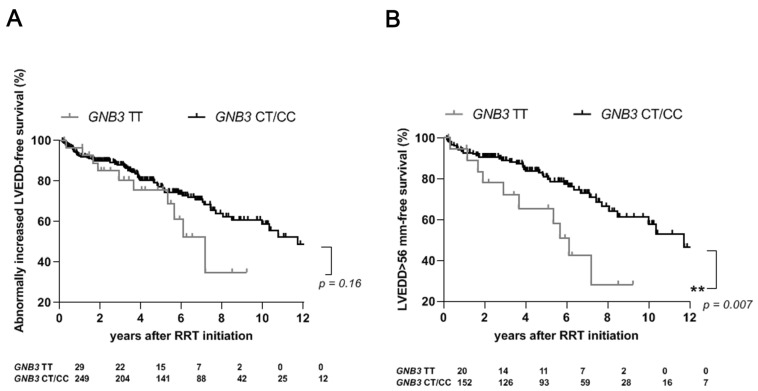
Development of abnormally elevated left ventricular end-diastolic diameter (LVEDD) values among carriers of the TT genotype and carriers of the CT/CC genotype of *GNB3* during the follow-up period of renal replacement therapy. (**A**) Survival dependent on the occurrence of abnormally increased LVEDD values among all 454 dialysis patients according to the genotype of *GNB3* (*p* = 0.16). (**B**) Survival dependent on the occurrence of abnormally increased LVEDD values according to the genotype of *GNB3*, considering only male dialysis patients (*p* = 0.007). **, *p* = 0.01. LVEDD, left ventricular end-diastolic diameter; RRT, renal replacement therapy.

**Table 1 ijms-24-15260-t001:** Baseline characteristics of 454 patients undergoing renal replacement therapy (RRT). CI, confidence interval; HUS, hemolytic uremic syndrome; RR, relative risk.

				RR (95% CI)	*p* Value			RR (95% CI)	*p* Value			RR (95% CI)	*p* Value
Demographic Characteristics	All Patients *n* = 454	*GNB3* TT *n* = 45	*GNB3* CT/CC *n* = 409			*GNAQ* TT/TT *n* = 97	*GNAQ* GC/TT/GC/GC *n* = 357			*GNAS* TT *n* = 118	*GNAS* CT/CC *n* = 336		
Age in years, median (range)	51 (18–81)	47 (22–70)	52 (18–63)		0.08	53 (21–75)	50 (18–81)		0.14	50 (21–75)	51 (18–81)		0.19
Women, n (%)	168 (37)	16 (36)	152 (37)	0.96 (0.6–1.4)	0.83	45 (46)	123 (34)	1.35 (1.0–1.7)	**0.03**	40 (34)	128 (38)	0.89 (0.7–1.2)	0.42
Men, n (%)	286 (63)	29 (64)	257 (63)	1.03 (0.8–1.3)	0.83	52 (54)	234 (66)	0.82 (0.7–1.0)	**0.03**	78 (66)	208 (62)	1.07 (0.9–1.2)	0.42
Previous renal transplants, n (%)	72 (16)	6 (13)	66 (16)	0.83 (0.4–1.7)	0.63	15 (15)	57 (16)	0.97 (0.6–1.6)	0.90	19 (16)	53 (16)	1.0 (0.6–1.6)	0.93
Time on dialysis (days), median (range)	1637 (50–7068)	1541 (82–5135)	1649 (50–7068)		0.36	1586 (85–4581)	1650 (50–7068)		0.72	1691 (114–7068)	1627 (50–6243)		0.43
Peritoneal dialysis patients, n (%)	96 (21)	9 (20)	87 (21)	0.94 (0.5–1.7)	0.84	27 (28)	69 (19)	1.44 (0.97–2.)	0.07	27 (23)	69 (21)	1.10 (0.8–1.6)	0.59
Obesity (grade 1–3), n (%)	115 (25)	13 (29)	102 (25)	1.16 (0.7–1.8)	0.56	31 (32)	84 (24)	1.36 (0.9–1.9)	0.09	31 (32)	84 (24)	1.40 (0.9–1.9)	0.09
Diabetes mellitus type I, n (%)	13 (3)	2 (4)	11 (3)	1.65 (0.4–6.2)	0.50	3 (3)	10 (3)	1.10 (0.3–3.6)	0.88	4 (3)	9 (3)	1.30 (0.4–3.8)	0.69
Development of diabetic glomerulosclerosis among type I diabetics, n (%)	12 (92)	2 (100)	10 (91)	1.10 (0.4–10.6)	0.66	3 (100)	9 (90)	1.10 (0.5–4.8)	0.57	4 (100)	8 (89)	1.13 (0.6–3.4)	0.49
Diabetes mellitus type II, n (%)	55 (12)	6 (13)	49 (12)	1.11 (0.5–2.3)	0.79	8 (8)	47 (13)	0.63 (0.3–1.2)	0.19	10 (8)	45 (13)	0.63 (0.3–1.2)	0.16
insulin-dependent diabetes mellitus type II, n (%)	19 (4)	2 (4)	17 (4)	1.07 (0.3–3.9)	0.93	6 (6)	13 (4)	1.70 (0.7–4.2)	0.27	5 (3)	15 (4)	0.76 (0.3–2.1)	0.62
Development of diabetic glomerulosclerosis among type II diabetics, n (%)	14 (25)	3 (50)	11 (22)	2.23 (0.8–4.9)	0.14	2 (25)	12 (26)	0.98 (0.3–2.8)	0.98	2 (20)	12 (27)	0.75 (0.2–2.3)	0.66
Diabetes mellitus type I-III, n (%)	69 (15)	8 (18)	61 (15)	1.19 (0.6–2.2)	0.61	11 (11)	58 (16)	0.70 (0.4–1.2)	0.23	14 (12)	55 (16)	0.72 (0.4–1.2)	0.24
Development of diabetic glomerulosclerosis among type I-III diabetics, n (%)	27 (39)	5 (63)	22 (36)	1.73 (0.8–2.9)	0.15	5 (45)	22 (38)	1.20 (0.5–2.2)	0.64	6 (43)	21 (38)	1.12 (0.5–2.1)	0.75
Therapy with lipid-lowering agents, n (%)	130 (29)	17 (38)	113 (28)	1.37 (0.9–2.0)	0.15	32 (33)	98 (27)	1.20 (0.9–1.7)	0.29	30 (25)	100 (30)	0.85 (0.6–1.2)	0.37
Antihypertensive therapy with ≥5 drugs, n (%)	42 (10)	4 (9)	38 (10)	0.98 (0.4–2.4)	0.96	9 (9)	33 (10)	0.98 (0.5–1.9)	0.96	9 (8)	33 (10)	0.79 (0.4–1.6)	0.52
**Causes of renal failure**													
1. Diabetic glomerulosclerosis, n (%)	27 (6)	5 (11)	22 (5)	2.10 (0.8–4.9)	0.12	5 (11)	22 (5)	0.85 (0.3–2.1)	0.71	6 (5)	21 (6)	0.81 (0.3–1.9)	0.65
due to diabetes mellitus type I, n (%)	12 (3)	2 (4)	10 (2)	1.80 (0.5–6.9)	0.43	3 (3)	9 (3)	1.23 (0.4–4.1)	0.76	4 (3)	8 (2)	1.42 (0.5–4.3)	0.56
due to diabetes mellitus type II, n (%)	14 (3)	3 (7)	11 (3)	2.50 (0.8–7.7)	0.14	2 (2)	12 (3)	0.61 (0.2–2.4)	0.51	2 (2)	12 (4)	0.48 (0.1–1.8)	0.31
2. Chronic glomerulonephritis, n (%)	161 (35)	18 (40)	143 (35)	1.14 (0.8–1.6)	0.50	40 (41)	121 (34)	1.22 (0.9–1.6)	0.18	46 (39)	115 (34)	1.14 (0.9–1.5)	0.35
3. Nephrosclerosis, n (%)	53 (12)	5 (11)	48 (12)	0.95 (0.4–2.1)	0.90	10 (3)	43 (12)	0.86 (0.4–1.6)	0.64	10 (8)	43 (13)	0.66 (0.3–1.2)	0.21
4. Polycystic kidney disease, n (%)	60 (13)	4 (9)	56 (14)	0.65 (0.3–1.6)	0.37	13 (13)	47 (13)	1.02 (0.6–1.8)	0.95	17 (14)	43 (13)	1.13 (0.7–1.9)	0.66
5. Tubulointerstitial nephritis, n (%)	15 (3)	3 (7)	12 (3)	2.27 (0.7–7.0)	0.18	5 (5)	10 (3)	1.84 (0.6–5.0)	0.25	4 (3)	11 (3)	1.04 (0.4–3.0)	0.95
6. Congenital anomalies, n (%)	32 (7)	3 (7)	29 (7)	0.94 (0.3–2.7)	0.92	4 (4)	28 (8)	0.53 (0.2–1.4)	0.20	9 (8)	23 (7)	1.11 (0.5–2.3)	0.78
7. Autoimmune disease, n (%)	13 (3)	1 (2)	12 (3)	0.76 (0.1–4.3)	0.79	5 (5)	8 (2)	2.30 (0.8–6.5)	0.13	6 (5)	7 (2)	2.44 (0.9–6.8)	0.09
8. Amyloidosis, n (%)	6 (1)	0 (0)	6 (1)	0.0 (0.0–5.5)	0.41	3 (3)	3 (1)	3.68 (0.9–15.7)	0.09	1 (1)	5 (1)	0.57 (0.1–3.6)	0,60
9. Reflux nephropathy/recurrent pyelonephritis, n (%)	25 (6)	0 (0)	25 (6)	0.0 (0.0–1.3)	0.09	4 (4)	21 (6)	0.70 (0.3–1.9)	0.50	9 (8)	16 (5)	1.60 (0.7–3.4)	0.24
10. HUS, n (%)	7 (2)	1 (2)	6 (1)	1.51 (0.2–9.1)	0.70	1 (1)	6 (2)	0.61 (0.1–3.8)	0.65	1 (1)	6 (2)	0.48 (0.1–2.9)	0.48
11. Other, n (%)	55 (12)	5 (11)	55 (13)	0.91 (0.4–2.0)	0.83	7 (7)	48 (13)	0.54 (0.3–1.1)	0.10	9 (8)	46 (14)	0.56 (0.3–1.1)	0.08

**Table 2 ijms-24-15260-t002:** Occurrence of cardiovascular events and certain echocardiographic variables of the left ventricle among patients undergoing renal replacement therapy in relation to the three polymorphisms in genes coding for the G protein subunits *GNB3* c.825C > T, *GNAQ* −695/−694GC > TT, and *GNAS* c.393C > T. CI, confidence interval; IVSd, interventricular septal end diastole; LVEDD, left ventricular end-diastolic diameter; LVPWd, left ventricular posterior wall end diastole; MACE, major adverse cardiovascular events; RR, relative risk; PAOD, peripheral artery occlusive disease.

				RR (95% CI)	*p* Value	*p* Value after Bonferroni Correction	*p* Value after Benjamini and Hochberg Correction			RR (95% CI)	*p* value	*p* Value after Bonferroni Correction	*p* Value after Benjamini and Hochberg Correction			RR (95% CI)	*p* value	*p* Value after Bonferroni Correction	*p* Value after Benjamini and Hochberg Correction
Cardiovascular Events During Dialysis	All Patients *n* = 454	*GNB3* TT *n* = 45	*GNB3* CT/CC *n* = 409					*GNAQ* TT/TT *n* = 97	*GNAQ* GC/TT/GC/GC *n* = 357					*GNAS* TT *n* = 118	*GNAS* CT/CC *n* = 336				
Cardiovascular event, n (%)	69 (15)	5 (11)	64 (16)	0.71 (0.3–1.6)	0.42	1.0	0.79	20 (21)	49 (14)	1.50 (0.9–2.4)	0.09	1.0	0.38	19 (16)	50 (15)	1.08 (0.7–1.7)	0.75	1.0	0.89
Cardiovascular event involving >1 organ system, n (%)	14 (3)	1 (2)	13 (3)	0.70 (0.1–3.9)	0.73	1.0	0.84	6 (6)	8 (2)	2.76 (1.0–7.4)	**0.05**	0.85	0.38	3 (3)	11 (3)	0.78 (0.2–2.5)	0.69	1.0	0.89
Acute PAOD, n (%)	18 (4)	2 (4)	16 (4)	1.14 (0.3–4.1)	0.86	1.0	0.86	6 (6)	12 (3)	1.84 (0.7–4.6)	0.21	1.0	0.62	6 (5)	12 (4)	1.42 (0.6–3.6)	0.47	1.0	0.87
Transient ischemic attack, n (%)	4 (1)	1 (2)	3 (1)	3.03 (0.4–20.4)	0.31	1.0	0.79	1 (1)	3 (1)	1.23 (0.2–8.4)	0.86	1.0	0.94	2 (2)	2 (1)	2.85 (0.5–16.0)	0.27	1.0	0.87
Stroke, n (%)	13 (3)	1 (2)	12 (3)	0.76 (0.1–4.3)	0.79	1.0	0.84	3 (3)	10 (3)	1.10 (0.3–3.6)	0.88	1.0	0.94	4 (3)	9 (3)	1.27 (0.4–3.8)	0.69	1.0	0.89
Carotid artery stenosis, n (%)	4 (1)	0 (0)	4 (1)	0.0 (0.0–8.4)	0.51	1.0	0.84	0 (0)	4 (1)	0.0 (0.0–3.5)	0.30	1.0	0.70	1 (1)	3 (1)	1.0 (0.1–6.5)	0.96	1.0	0.96
Myocardial infarction, n (%)	25 (5)	0 (0)	25 (6)	0.0 (0.0–1.3)	0.09	1.0	0.68	7 (7)	18 (5)	1.43 (0.6–3.2)	0.41	1.0	0.70	5 (4)	20 (6)	0.71 (0.3–1.8)	0.48	1.0	0.87
Coronary artery stenosis requiring stent insertion, n (%)	34 (7)	1 (2)	33 (8)	0.28 (0.05–1.5)	0.16	1.0	0.68	14 (14)	20 (6)	2.58 (1.4–4.8)	**0.003**	**0.05**	**0.05**	7 (6)	27 (8)	0.74 (0.3–1.6)	0.46	1.0	0.87
Heart valve intervention, n (%)	16 (4)	1 (2)	15 (4)	0.61 (0.1–3.4)	0.62	1.0	0.84	4 (4)	12 (3)	1.23 (0.4–3.5)	0.72	1.0	0.94	7 (6)	9 (3)	2.21 (0.9–5.6)	0.10	1.0	0.87
Coronary artery bypass grafting, n (%)	14 (3)	1 (2)	13 (3)	0.70 (0.1–3.9)	0.73	1.0	0.84	3 (3)	11 (3)	1.0 (0.3–3.2)	0.99	1.0	0.99	5 (4)	9 (3)	1.58 (0.6–4.4)	0.40	1.0	0.87
MACE (cardiovascular death, stroke, myocardial infarction), n (%)	36 (8)	1 (2)	35 (9)	0.26 (0.1–1.4)	0.14	1.0	0.68	10 (10)	26 (7)	1.42 (0.7–2.8)	0.33	1.0	0.70	8 (7)	28 (8)	0.81 (0.4–1.7)	0.59	1.0	0.89
**New onset of chronic cardiovascular disease during dialysis**																			
Coronary artery disease, n (%)	60 (13)	5 (11)	55 (13)	0.83 (0.3–1.8)	0.66	1.0	0,84	18 (19)	42 (12)	1.58 (0.9–2.6)	0.08	1.0	0.38	13 (11)	47 (14	0.79 (0.4–1.4)	0.41	1.0	0.87
Coronary artery disease of ≥2 vessels, n (%)	29 (6)	0 (0)	29 (7)	0.0 (0.0–1.1)	0.07	1.0	0.68	8 (8)	21 (6)	1.40 (0.6–3.0)	0.40	1.0	0.70	7 (6)	22 (7)	0.91 (0.4–2.0)	0.81	1.0	0.89
Heart failure, n (%)	68 (15)	6 (13)	62 (15)	0.88 (0.4–1.8)	0.75	1.0	0.84	13 (13)	55 (15)	0.87 (0.5–1.5)	0.62	1.0	0.88	17 (14)	51 (15)	0.95 (0.6–1.6)	0.84	1.0	0.89
**Echocardiographic parameters of left ventricle**																			
LVEDD male > 56 mm/female > 51 mm (278/454), n (%)	78 (28)	10 (34)	68 (27)	1.26 (0.7–2.0)	0.42	1.0	0.79	17 (29)	61 (28)	1.06 (0.7–1.6)	0.81	1.0	0.94	17 (23)	61 (30)	0.77 (0.5–1.2)	0.26	1.0	0.87
IVSd male > 12 mm/female > 11 mm (299/454), n (%)	146 (49)	11 (41)	135 (50)	0.82 (0.5–1.2)	0.38	1.0	0.79	33 (56)	113 (47)	1.19 (0.9–1.5)	0.22	1.0	0.62	37 (46)	109 (50)	0.91 (0.7–1.2)	0.51	1.0	0.87
LVPWd male 12 mm/female > 12 mm (208/454), n (%)	74 (36)	10 (45)	64 (34)	1.32 (0.8–2.0)	0.31	1.0	0.79	13 (32)	61 (37)	0.87 (0.5–1.4)	0.56	1.0	0.87	19 (32)	55 (37)	0.85 (0.5–1.3)	0.45	1.0	0.87

**Table 3 ijms-24-15260-t003:** Results of univariate analysis and multivariable Cox regression identifying risk factors for coronary artery stenosis requiring stent insertion among 454 patients undergoing renal replacement therapy. CI, confidence interval; HR, hazard ratio; LVEDD, left ventricular end-diastolic diameter; RR, relative risk.

					Multivariable Cox regression considering only significantly different variables of the univariate analysis			Multivariable Cox regression considering all variables		
			Univariate RR (95% CI)	*p* Value	HR	95% CI	*p* Value	HR	95% CI	*p* Value
Variable	Coronary Artery Stenosis Requiring Stent Insertion *n* = 34	No Coronary Artery Stenosis *n* = 420								
Age in years, median	59	50		**<0.0001**	1.07	1.0–1.1	**0.001**	1.07	1.0–1.1	**0.001**
Males, n (%)	27 (79)	259 (62)	1.29 (1.0–1.5)	**0.04**	1.89	0.8–4.6	0.16	2.00	0.8–4.9	0.13
Previous renal transplants, n (%)	8 (24)	64 (15)	1.54 (0.8–2.8)	0.20				1.23	0.4–3.5	0.69
Time on dialysis (days), median	2424	1596		**0.001**	1.00	1.0–1.0	0.92	1.00	1.0–1.0	0.98
Peritoneal dialysis patients, n (%)	4 (12)	92 (22)	0.54 (0.2–1.2)	0.16				1.03	0.3–3.3	0.96
Obesity, n (%)	10 (29)	105 (25)	1.18 (0.7–1.9)	0.57				0.58	0.2–1.6	0.28
Diabetes mellitus type I-III, n (%)	11 (32)	58 (14)	2.34 (1.3–3.8)	**0.004**	4.74	2.0–11.3	**0.0005**	6.08	2.3–15.9	**0.0002**
LVEDD male > 56 mm/female > 51 mm (278/454), n (%)	12 (44)	66 (26)	1.69 (1.0–2.6)	**0.05**	2.06	1.0–4.5	0.07	2.14	0.9–4.9	0.07
*GNAQ* TT/TT genotype	14 (41)	83 (20)	2.08 (1.3–3.1)	**0.003**	4.29	1.8–10.1	**0.001**	4.46	1.9–10.5	**0.001**

**Table 4 ijms-24-15260-t004:** Results of univariate analysis and multivariable Cox regression identifying risk factors for the development of cardiovascular events among 454 patients undergoing renal replacement therapy. CI, confidence interval; HR, hazard ratio; LVEDD, left ventricular end-diastolic diameter; RR, relative risk.

					Multivariable Cox Regression Considering Only Significantly Different Variables of the Univariate Analysis			Multivariable Cox Regression Considering All Variables		
			Univariate RR (95% CI)	*p* Value	HR	95% CI	*p* Value	HR	95% CI	*p* Value
Variable	Cardiovascular Event *n* = 69	No Events *n* = 385								
Age in years, median	57	49		**<0.0001**	1.05	1.0–1.1	**0.005**	1.05	1.0–1.1	**0.001**
Males, n (%)	51 (74)	235 (61)	1.21 (1.0–1.4)	**0.04**	1.60	0.9–2.9	0.13	1.71	0.9–3.2	0.09
Previous renal transplants, n (%)	14 (20)	58 (15)	1.35 (0.8–2.2)	0.27				1.57	0.8–3.2	0.21
Time on dialysis (days), median	2260	1551		**<0.0001**	1.00	1.0–1.0	0.23	1.00	1.0–1.0	0.27
Peritoneal dialysis patients, n (%)	14 (20)	82 (21)	0.95 (0.6–1.5)	0.85				1.55	0.8–3.2	0.24
Obesity, n (%)	20 (29)	95 (25)	1.17 (0.8–1.7)	0.45				0.78	0.4–1.6	0.47
Diabetes mellitus type I-III, n (%)	23 (33)	46 (12)	2.79 (1.8–4.2)	**<0.0001**	3.16	1.7–5.8	**0.002**	3.79	1.9–7.5	**0.0001**
LVEDD male > 56 mm/female > 51 mm (278/454), n (%)	20 (38)	58 (26)	1.5 (0.98–2.2)	**0.06**	1.52	0.9–2.7	0.15	1.54	0.9–2.8	0.15
*GNAQ* TT/TT genotype	20 (29)	77 (20)	1.45 (0.9–2.2)	**0.09**	1.93	1.0–3.6	**0.04**	1.93	1.0–3.6	**0.04**

**Table 5 ijms-24-15260-t005:** Results of univariate analysis and multivariable Cox regression identifying risk factors for the occurrence of cardiovascular events affecting multiple organ systems among 454 patients undergoing renal replacement therapy. CI, confidence interval; HR, hazard ratio; LVEDD, left ventricular end-diastolic diameter; RR, relative risk.

					Multivariable Cox Regression Considering Only Significantly Different Variables of the Univariate Analysis			Multivariable Cox Regression Considering All Variables		
			Univariate RR (95% CI)	*p* Value	HR	95% CI	*p* Value	HR	95% CI	*p* Value
Variable	Cardiovascular Events Involving >1 Organ System *n* = 14	Without Involvement of >1 Organ System *n* = 440								
Age in years, median	58	51		**0.005**	1.07	1.0–1.2	**0.06**	1.10	1.0–1.2	**0.04**
Males, n (%)	10 (71)	276 (63)	1.14 (0.7–1.4)	0.51				0.65	0.2–2.5	0.53
Previous renal transplants, n (%)	5 (36)	67 (15)	2.35 (1.1–4.3)	**0.04**	1.45	0.4–5.4	0.58	2.71	0.6–13.5	0.22
Time on dialysis (days), median	3505	1596		**<0.0001**	1.00	1.0–1.0	**0.001**	1.00	1.0–1.0	**0.002**
Peritoneal dialysis patients, n (%)	1 (7)	95 (22)	0.33 (0.1–1.5)	0.19				0.00	0.0–10.1	0.97
Obesity, n (%)	6 (43)	109 (25)	1.73 (0.8–2.8)	0.13				5.73	1.3–25.9	**0.02**
Diabetes mellitus type I-III, n (%)	4 (29)	65 (15)	1.93 (0.8–3.9)	0.16				2.87	0.5–15.4	0.22
LVEDD male >56 mm/female > 51 mm(278/454), n (%)	7 (64)	71 (27)	2.39 (1.3–3.5)	**0.007**	4.50	1.2–17.2	**0.03**	4.25	1.0–19.1	0.06
*GNAQ* TT/TT genotype	6 (43)	91 (21)	2.07 (1.0- 3.4)	**0.046**	4.77	1.4–16.8	**0.02**	4.91	1.2–19.9	**0.03**

**Table 6 ijms-24-15260-t006:** Occurrence of cardiovascular events and certain echocardiographic variables of the left ventricle among male and female patients undergoing renal replacement therapy in relation to the *GNB3* genotype. CI, confidence interval; IVSd, interventricular septal end diastole; LVEDD, left ventricular end-diastolic diameter; LVPWd, left ventricular posterior wall end diastole; MACE, major adverse cardiovascular events; RR, relative risk; PAOD, peripheral artery occlusive disease.

				RR (95% CI)	*p* Value	*p* Value after Bonferroni Correction	*p* Value after Benjamini and Hochberg Correction				RR (95% CI)	*p* Value	*p* Value after Bonferroni Correction	*p* Value after Benjamini and Hochberg correction
Cardiovascular Events During Dialysis	All Men *n* = 286	*GNB3* TT *n* = 29	*GNB3* CT/CC *n* = 257					All Women *n* = 168	*GNB3* TT *n* = 16	*GNB3* CT/CC *n* = 152				
Cardiovascular event, n (%)	51 (18)	4 (14)	47 (18)	0.75 (0.3–1.8)	0.55	1.0	0.87	18 (11)	1 (6)	17 (11)	0.56 (0.1–2.7)	0.54	1.0	0.79
Cardiovascular event involving >1 organ system, n (%)	10 (3)	1 (3)	9 (4)	0.99 (0.2–5.5)	0.99	1.0	0.99	4 (2)	0 (0)	4 (3)	0.0 (0.0–8.1)	0.51	1.0	0.79
Acute PAOD, n (%)	13 (5)	2 (7)	11 (4)	1.61 (0.4–5.9)	0.52	1.0	0.87	5 (3)	0 (0)	5 (3)	0.0 (0.0–6.4)	0.46	1.0	0.79
Transient ischemic attack, n (%)	1 (0.3)	0 (0)	1 (0.4)	0.0 (0.0–33.1)	0.74	1.0	0.94	3 (2)	1 (6)	2 (1)	4.75 (0.6–33.7)	0.16	1.0	0.79
Stroke, n (%)	8 (3)	1 (3)	7 (3)	1.11 (0.2–6.3)	0.92	1.0	0.98	4 (2)	0 (0)	4 (3)	0.0 (0.0–8.1)	0.51	1.0	0.79
Carotid artery stenosis, n (%)	3 (1)	0 (0)	3 (1)	0.0 (0.0–10.7)	0.56	1.0	0.87	1 (1)	0 (0)	1 (1)	0.0 (0.0–34.8)	0.75	1.0	0.79
Myocardial infarction, n (%)	19 (7)	0 (0)	19 (7)	0.0 (0.0–1.6)	0.13	1.0	0.68	6 (4)	0 (0)	6 (4)	0.0 (0.0–5.3)	0.42	1.0	0.79
Coronary artery stenosis requiring stent insertion, n (%)	27 (9)	1 (3)	26 (10)	0.34 (0.1–1.8)	0.24	1.0	0.68	7 (4)	0 (0)	7 (5)	0.0 (0.0–4.5)	0.38	1.0	0.79
Heart valve intervention, n (%)	13 (5)	1 (3)	12 (5)	0.74 (0.1–4.0)	0.77	1.0	0.94	3 (2)	0 (0)	3 (2)	0.0 (0.0–11.0)	0.57	1.0	0.79
Coronary artery bypass grafting, n (%)	12 (4)	1 (3)	11 (4)	0.81 (0.1–4.4)	0.83	1.0	0.94	2 (1)	0 (0)	2 (1)	0.0 (0.0–16.9)	0.64	1.0	0.79
MACE (cardiovasacular death, stroke, myocardial infarction), n (%)	26 (9)	1 (4)	25 (10)	0.35 (0.1–1.8)	0.26	1.0	0.68	10 (6)	0 (0)	10 (7)	0.0 (0.0–3.1)	0.29	1.0	0.79
**New onset of chronic cardiovascular disease during dialysis**														
Coronary artery disease, n (%)	46 (16)	4 (14)	42 (16)	0.84 (0.3–2.0)	0.72	1.0	0.93	14 (8)	1 (6)	13 (9)	0.73 (0.1–3.7)	0.75	1.0	0.79
Coronary artery disease of ≥2 vessels, n (%)	24 (8)	0 (0)	24 (9)	0.0 (0.0–1.3)	0.09	1.0	0.68	5 (3)	0 (0)	5 (3)	0.0 (0.0–6.4)	0.46	1.0	0.79
Heart failure, n (%)	52 (18)	3 (10)	49 (19)	0.54 (0.2–1.4)	0.25	1.0	0.68	16 (10)	3 (19)	13 (9)	2.19 (0.7–6.0)	0.19	1.0	0.79
**Echocardiographic parameters of left ventricle**														
LVEDD male >56 mm/female > 51 mm(278/454), n (%)	49 (17)	10 (50)	39 (26)	1.95 (1.1–3.1)	**0.02**	0.32	0.34	29 (27)	0 (0)	29 (30)	0.0 (0.0–1.0)	0.05	0.80	0.79
IVSd male > 12 mm/female > 11 mm (299/454), n (%)	98 (34)	7 (41)	91 (51)	0.80 (0.4–1.3)	0.42	1.0	0.87	48 (46)	4 (40)	44 (46)	0.86 (0.4–1.6)	0.70	1.0	0.79
LVPWd male 12 mm/female > 12 mm (208/454), n (%)	58 (20)	8 (57)	50 (42)	1.36 (0.8–2.0)	0.28	1.0	0.68	16 (8)	2 (25)	14 (21)	1.20 (0.3–3.4)	0.79	1.0	0.79

**Table 7 ijms-24-15260-t007:** Results of univariate analysis and multivariable Cox regression identifying risk factors for the development of abnormally increased LVEDD values among 286 male patients undergoing renal replacement therapy. CI, confidence interval; HR, hazard ratio; LVEDD, left ventricular end-diastolic diameter; RR, relative risk.

					Multivariable Cox Regression Considering All Variables		
			Univariate RR (95% CI)	*p* Value	HR	95% CI	*p* Value
Variable	LVEDD > 56 mm *n* = 49	LVEDD ≤ 56 mm *n* = 123					
Age in years, median	54	49		0.19	1.01	1.0–1.0	0.44
Previous renal transplants, n (%)	8 (16)	18 (15)	1.12 (0.5–2.3)	0.78	0.90	0.4–2.0	0.79
Time on dialysis (days), median	1488	1834		0.17	1.00	1.0–1.0	**0.00001**
Peritoneal dialysis patients, n (%)	8 (16)	24 (20)	0.84 (0.4–1.7)	0.63	0.93	0.4–2.1	0.86
Obesity, n (%)	15 (31)	32 (26)	1.18 (0.7–1.9)	0.54	1.22	0.6–2.4	0.57
Diabetes mellitus type I–III, n (%)	8 (16)	17 (14)	1.18 (0.5–2.5)	0.67	1.05	0.4–2.5	0.92
Coronary artery disease, n (%)	10 (20)	22 (18)	1.14 (0.6–2.2)	0.70	0.92	0.4–2.0	0.83
*GNB3* TT genotype	10 (20)	10 (8)	2.51 (1.1–5.5)	**0.02**	2.35	1.2–4.8	**0.02**

**Table 8 ijms-24-15260-t008:** Adjustment of antihypertensive drug therapy on the follow-up of RRT among all 454 patients undergoing renal replacement therapy and in the subgroups of male and female patients undergoing renal replacement therapy in relation to the *GNB3* genotype. CI, confidence interval; RR, relative risk; RRT, renal replacement therapy.

				RR (95% CI)	*p* Value				RR (95% CI)	*p* Value				RR (95% CI)	*p* Value
	All Patients *n* = 435	*GNB3* TT *n* = 43	*GNB3* CT/CC *n* = 392			All Men *n* = 274	*GNB3* TT *n* = 29	*GNB3* CT/CC *n* = 245			All Women *n* = 161	*GNB3* TT *n* = 14	*GNB3* CT/CC *n* = 147		
Reduction of antihypertensive therapy from ≥5 to ≤5 drugs, n (%)	25 (6)	2 (5)	23 (6)	0.79 (0.2–2.8)	0.75	16 (6)	2 (7)	14 (6)	1.21 (0.3–4.3)	0.8	9 (6)	0 (0)	9 (6)	0.0 (0.0–3.8)	0.34
No change in antihypertensive therapy with ≥5 drugs, n (%)	17 (4)	2 (5)	15 (4)	1.22 (0.3–4.5)	0.79	13 (5)	1 (3)	12 (5)	0.71 (0.1–3.9)	0.73	4 (2)	1 (7)	3 (2)	3.5 (0.5–22.0)	0.24
Enchantment of antihypertensive therapy from ≤5 to ≥5 drugs, n (%)	45 (10)	9 (21)	36 (9)	2.3 (1.2–4.2)	**0.02**	27 (10)	2 (7)	25 (10)	0.68 (0.2–2.3)	0.57	18 (11)	7 (50)	11 (7)	6.68 (3.0–13.7)	**0.0001**
No change in antihypertensive therapy with ≤5 drugs, n (%)	348 (80)	30 (70)	318 (81)	0.86 (0.7–1.0)	0.08	218 (80)	24 (83)	194 (79)	1.05 (0.8–1.2)	0.65	130 (81)	6 (43)	124 (84)	0.51 (0.3–0.8)	**0.0002**

**Table 9 ijms-24-15260-t009:** Information about primers, annealing temperatures, and restriction enzymes that were used for the genotyping of *GNAQ, GNB3,* and *GNAS*.

Gene	rsID	Position	5′-3′Primer	3′-5′ Primer	Annealing Temperature (°C)	Restriction Enzyme
*GNAS*	rs7121	c.393C > T	TGT GGC CGC CAT GAG CAA	TAA GGC CAC ACA AGT CGG GGT	64 °C	BseGI
*GNAQ*	rs72466452	g.78032143delinsAA	AGG GTG CGG GAG CAG TAG GCG	CCT CCT GGA AGG CTT TCC TGG G	62 °C	PdiI
*GNB3*	rs5443	c.825C > T	GCT GCC CAG GTC TGA TCC C	TGG GGA GGG TCC TTC CAG C	66 °C	BseDI

## Data Availability

The data presented in this study are available on request from the corresponding author. The data are not publicly available because of restrictions, e.g., privacy or ethics.

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
