# Peer review of "Single-Nucleotide Polymorphism in Genes Encoding G Protein Subunits GNB3 and GNAQ Increase the Risk of Cardiovascular Morbidity among Patients Undergoing Renal Replacement Therapy"

_ijms, 2023, doi:10.3390/ijms242015260_

Round 1
Reviewer 1 Report
This study attempts to find associations between polymorphisms in 3 G protein subunits and cardiovascular events. I think that there are major flaws in this work that make the conclusions unsafe.
1. Seeking associations between multiple polymorphisms (9?) and multiple CV events in a fairly small number of patients allows a very high risk of statistical associations occurring by random chance.
2. This problem is exacerbated when the polymorphisms are analysed with effects of sex differences giving perhaps 18 groups.
2. No allowance has been made for this problem by attempting statistical corrections for multiple comparisons (eg Bonferroni correction). Even with such correction however, I doubt that this study can provide data from which safe conclusions can be drawn.
3. The term 'acute cardiovascular events' is never clearly defined and seems to include many events that would not usually be included such as coronary stent placement and CABG which are often performed as elective procedures in patients with stable disease. The onset of 'newly diagnosed coronary artery disease', development of LVH and many other problem such as new onset heart failure (hard to diagnose in RRT patients) are also not conventionally included. A strict diagnosis of MACE includes CV death, acute stroke or myocardial infarction only. Hospitalisation for heart failure is sometimes added.
4. LVH is defined by an increase in LV mass (usually indexed for BSA) so cannot be used as an an endpoint unless this has been measured. LV wall thickness can be used as a surrogate but has problems. LVEDD is a measure of LV dilatation and not hypertrophy.
5. The use of a population listed for transplant excludes many high risk patients who will have been screened out prior to listing decisions.
6. Did the analysis combine prevalent and incident CVD? This requires explanation.
Reviewer 2 Report
inappropriate number of self-citations by author W. Siffert
|
|
Reviewer 3 Report
Usually genotype is understood as a combination of 2 alleles (maternal and paternal - CC, CT, TT). TT/TT, GC/TT, GC/GC would more correctly be called a combination of genotypes.
It is not clear why patients of very different ages from 18 to 81 years old are analyzed in one group? Do the authors assume that diseases and their complications in patients of different ages have the same course and the probability of complications is also similar? This is an inadequate approach to analyzing this kind of data and will lead to erroneous conclusions. It is necessary to divide into acceptable age groups for analyzing age-related diseases.
Table #1 shows the age in carriers of different TT, ST/TT genotypes. What is the purpose of this information? How to interpret it?
Table #1 shows the frequencies of genotypes in women. Why aren't they listed in men?
It is mechanically easier to count the percentages of Diabetic glomerulosclerosis in relation to the whole group, but it was more interesting to see the percentages of the population of patients with diabetes. The same goes for renal failure on the background of due to diabetes mellitus type I and due to diabetes mellitus type II.
Why does the text repeat the information that is presented in Table No. 1? "With respect to the distribution of the three relevant single-nucleotide polymor-117 phisms encoding G protein subunits, the TT genotype of GNB3was found in 45 patients 118 (10%), and the CT/CC genotypes were found in 409 (90%) (Table 1). We detected the 119 TT/TT genotype of GNAQin 97 patients (21%), whereas 357 (79%) carried the GC/TT or 120 GC/GC genotype (Table 1). Of the 454 patients in the cohort, 118 (26%) carried the TT." This does not provide the reader with any additional useful information.
The analysis presented in Table 2 also raises questions: how to interpret calculations made on the total group, where both men and women, young and old, are included?
It is not clear whether the authors checked whether the distribution of genotypes corresponds to the Hardy-Weinberg equilibrium or not. If they did, is it consistent or not? If not, it should be checked and stated so.
To what extent did the accuracy of relative risk estimates in multivariate models improve when genotypes were added to them?
An explicit reference to the very preliminary nature of the data obtained in multivariable models because of the very small size of the study groups should be added to the study limitations. For example, Table 3, Coronary artery stent insertion n=34 (very small for multivariate analysis); Table 4, Acute cardiovascular event n=69; Table 5, Acute cardiovascular event involving >1 organ system n=14 (does multivariate analysis even make sense with this size group?); Table 7, LVEDD >56 mm n=49.
Round 2
Reviewer 1 Report
I remain of the opinion that this paper does not provide data from which safe conclusions can be drawn.
The only end point that remains significant after multiple testing correction is coronary stenting and this is a procedure not an event. It is also indicated for multiple and very different reasons including persistent angina despite medical therapy (chronic CAD) and myocardial infarction (acute CAD). Putting all stenting events together increases numbers but does not make much sense from the point of view of defining CV pathological events.
NB. The table are a real presentational mess with misaligned columns and rows. Impossible to interpret. This is probably a journal editing issue rather than a fault of the authors
Minor improvements needed if the paper were to be published.
Reviewer 2 Report
The authors report interesting results on cardiovascular morbidity in patients with CKD-G5 and renal replacement therapy related to single-nucleotide polymorphisms with proposed functional significance for G-protein β3 subunit and G-protein αq subunit-dependent signal transduction.
Some issues need revision:
Major:
1. Correction for multiple testing should be performed according to Benjamini-Hochberg.
2. The variables in the multivariable regression analysis should be independent from each other.
3. Variables for the multivariable model should not be selected by significance in a univariate model. A non-significant predictor in a univariate analysis can be very significant in multivariate analysis.
Minor:
4. Table 1 does not have a heading.
5. In several of the tables the data cannot be read, e.g. p-values in Tables 1 and 2.
6. The Kaplan-Maier analyses are barely discernible.
Reviewer 3 Report
Accept in present form
Round 3
Reviewer 1 Report
The authors have made many changes that have improved the manuscript including acknowledging that some of these findings may be due to play of chance.
Needs some minor changes to improve the style and flow
Reviewer 2 Report
Thanks for your revised version of the manuscript.